# LACTB mRNA expression is increased in pancreatic adenocarcinoma and high expression indicates a poor prognosis

Jian Xie[1]⊙, Yang Peng[2]⊙, Xiaoyu Chen[3], Qigang Li[1], Bin Jian[1], Zelin Wen[1], Shengchun Liu●[2]*

1 Department of General Surgery, Yong Chuan Hospital of Chongqing Medical University, Chongqing, China, 2 Department of Endocrine and Breast Surgery, The First Affiliated Hospital of Chongqing Medical University, Chongqing, China, 3 Department of Prevention and Health Protection, Yong Chuan Hospital of Chongqing Medical University, Chongqing, China

⊙ These authors contributed equally to this work.
* liushengchun1968@163.com

**Data Availability Statement:** The expression data and target genes are available from the TCGA Research Network (http://cancergenome.nih.gov/)

## Abstract

This study aimed to find the prognostic value of Beta-lactamase-like (LACTB) in pancreatic adenocarcinoma (PAAD) patients. The mRNA expression of LACTB was upregulated in PAAD and was correlated with vital status (P = 0.0199). The immunoreactive scores of LACTB protein in human PAAD tissues were significantly higher than those in adjacent non-cancerous pancreatic tissues. Receiver operating characteristic (ROC) curve assessment showed that LACTB mRNA expression has high diagnostic value in PAAD. Kaplan-Meier curve and Cox analyses suggested that patients with high LACTB mRNA expression have a poor prognosis, indicating that LACTB mRNA is an independent prognostic factor for overall survival [hazard ratio (HR) = 1.72, P = 0.015, 95% confidence interval (CI) = 1.106–2.253] and disease-specific survival (HR = 1.97, P = 0.004, 95% CI = 1.238–3.152) of PAAD patients. Gene set enrichment analysis (GSEA) revealed that hallmark_g2m_checkpoint, hallmark_myc_targets_v1, hallmark_e2f_targets, and kegg_cell_cycle were differentially enriched in phenotypes with high LACTB expression. In addition, CDC20, CDK4, MCM6, MAD2L1, MCM2 and MCM5 were leading genes intersecting in these four pathways, and a positive correlation between mRNA expression and LACTB was observed in most normal and cancer tissues. Finally, elevated LACTB mRNA expression was significantly related to multiple immune marker sets. Our results elucidate that LACTB is involved in the development of cancer, and that high LACTB expression in patients with PAAD can predict a poor prognosis. High LACTB expression was significantly correlated with cell cycle-related genes and multiple immune marker sets.

## Introduction

Pancreatic adenocarcinoma (PAAD) is lethal and aggressive, with a low 5-year survival rate [1]. Despite extensive research and clinical advances, the mortality of pancreatic cancer is

and the GTEx program (https://www.gtexportal.
org/).

**Funding:** The present study was supported by
grants from the National Natural Science
Foundation of China (nos. 81772979 and
81472658) and Natural Sciences Foundation
project proposals of YongChuan (Ycstc,
2019nb0203).

**Competing interests:** The authors have declared
that no competing interests exist.

increasing, and it is estimated to become the second most common cause of cancer-related
deaths by 2030 [2]. One important reason is that only <20% of all patients are eligible for
resection, as most patients have evidence of distant metastasis at the time of diagnosis [3].
However, the advancement of new prediction tools based on prognosis-related genes has been
promoted by the development of tumor molecular biology. Some prognostic markers that
reflect tumor progression at the molecular level may help to achieve more accurate individual
survival prediction.

Metabolic dysregulation is critical to the progression of cancers, including PAAD. A disruption in glutamine metabolism can ultimately lead to the inhibition of PAAD growth in vitro
and in vivo [4].

Beta-lactamase-like (LACTB) is a mitochondrial protein that is associated evolutionarily
with bacterial penicillin-binding/beta-lactamase proteins [5]. LACTB has been shown to be
ubiquitous in different mammalian tissues, most notably in the skeletal muscle, heart and liver
[6]. LACTB acts as a new protease homologue and is involved in the regulation of metabolic
circuitry and cellular metabolic processes. Moreover, LACTB can regulate intramitochondrial
membrane organization and energy homeostasis [7]. More recently, it was reported that
LACTB is a tumor suppressor that inhibits the proliferation and promotes the apoptosis of
breast cancer cells [8]. Kaixuan Zeng et al. found that low LACTB expression was associated
with poor overall survival (OS) in colorectal cancer patients, and LACTB was also determined
to be an independent prognostic factor for poor outcomes [9]. Additionally, Chen Xue et al.
demonstrated that both LACTB mRNA and protein levels were downregulated in hepatocellular carcinoma and that low LACTB expression was associated with poor OS and relapse-free
survival [10]. However, to date, the expression of LACTB and its clinicopathological significance in PAAD have not been identified.

In this study, we demonstrated that the LACTB gene acts as an oncogene and that the
mRNA expression and protein expression of LACTB are significantly higher in PAAD
tumor tissue than in adjacent normal tissue. ROC curve assessment indicated that LACTB
mRNA expression has high diagnostic value in PAAD. Moreover, high expression of
LACTB mRNA is an independent prognostic factor for OS and disease-specific survival
(DSS) in patients with PAAD. Furthermore, high LACTB expression is correlated with cell
cycle-related genes and multiple immune marker sets. Gene set enrichment analysis (GSEA)
revealed that HALLMARK_G2M_CHECKPOINT, HALLMARK_MYC_TARGETS_V1,
HALLMARK_E2F_TARGETS, and HALLMARK_E2F_TARGETS were differentially
enriched in phenotypes with high LACTB expression, which may be important biological
pathways in the pathogenesis of pancreatic cancer and warrant further study.

## Materials and methods

### Data mining and collection

The expression data for LACTB and target genes were derived from the TCGA Research Network (http://cancergenome.nih.gov/) and the GTEx program (https://www.gtexportal.org/),
and four majors clinical endpoints were obtained from the Pan-Cancer Clinical Data Resource
(TCGA-CDR). Normalized gene expression data for the TCGA-PAAD dataset were log2-
transformed for further analysis. GSEA was performed with the clusterProfiler R/Bioconductor package [11]. GSEA generated an ordered list of all genes according to LACTB mRNA
expression using the GSEA hallmark gene set [12]. The functional gene set files "h.all.v7.1.
entrez.gmt" and "c2.cp.kegg.v7.1.entrez.gmt" were used. The nominal P-value and NES were
used to sort the pathways enriched in each phenotype.

## Immunohistochemical staining

For IHC analysis, a tissue microarray including 98 primary pancreatic cancer tissues and 68 noncancerous pancreatic tissues was obtained from Shanghai Outdo Biotech Co., Ltd. (Shanghai, People's Republic of China; Category no: HPan-Ade170Sur-01). All the samples were fully anonymized and none of the samples collected in this study received chemotherapy or radiotherapy before surgery. The expression patterns and subcellular localizations of LACTB proteins in clinical pancreatic cancer tissues were detected by IHC, and the immunoreactivity scores (IRSs) were calculated. Briefly, paraffin-embedded tissues were cut at 4 μm and then deparaffinized with xylene. Following simple proteolytic digestion and peroxidase blocking, the tissue slides were incubated overnight with a primary antibody against LACTB (18195-1-AP, Proteintech, Wuhan) at a dilution of 1:2000 at 4 ˚C. After washing, the peroxidase-labeled polymer and substrate–chromogen were then employed to visualize staining of the protein of interest. In each IHC run, negative controls were carried out by omitting the primary antibody. Following hematoxylin counterstaining, immunostaining was scored by two independent experienced pathologists who were blinded to the clinicopathological data and clinical outcomes of the patients. The scores of the two pathologists were compared, and any discrepant scores were trained by re-examining the staining by both pathologists to achieve a consensus score. The number of positively stained cells in 10 representative microscopic fields was counted, and the percentage of positive cells was calculated. Images of IHC staining were obtained and analyzed. LACTB was scored according to staining intensity from 1+ to 3+. A score of 1+ to 2+ was defined as low LACTB expression, whereas a score of 3+ was defined as high LACTB expression.

## Single-sample gene set enrichment analysis (ssGSEA)

The tumor-infiltrating fraction of diverse immune cell subtypes was calculated using ssGSEA in the gsva R package (Version 1.36.1, http://www.bioconductor.org/packages/release/bioc/html/GSVA.html). ssGSEA transforms specific gene expression patterns into quantities of immune cell populations in individual tumor patients. The cell marker was downloaded from a previous study [13] and used to evaluate differences in the tumor-infiltrating fractions of 28 human immune cell phenotypes between PAAD patients with distinct LACTB mRNA expression statuses.

## Statistical analysis

Statistical analysis was performed in R v. 3.4.3. The ggplot2 and ggsurvplot packages in the statistical software R were used to generate graphs. Discrete variables were represented by box plots to measure differences in expression. The chi-square test was performed to examine the clinical relationship between high and low LACTB mRNA expression patients. Kaplan-Meier curves indicated that clinicopathological traits were correlated with OS, DSS, DFI, and PFI. Univariate Cox regression analysis was applied to choose the variables of interest, followed by multivariate Cox regression to analyze the relationship between LACTB mRNA expression and OS rate in PAAD patients. The hazard ratio (HR) and 95% confidence interval (CI) were calculated to identify genes associated with OS. Unless otherwise stipulated, $P < 0.05$ was considered statistically significant. The cutoff value was determined by the best separation value of LACTB mRNA expression using the survival R package. $P < 0.05$ was considered statistically significant.

## Results

### Patient characteristics

To detect the clinical significance of LACTB expression in PAAD, we analyzed The Cancer Genome Atlas (TCGA) datasets, and the clinical and gene expression data for 183 cases of

primary pancreatic cancer were downloaded from the TCGA-PAAD dataset. As shown in Table 1, 68.31% of patients were older than 60 years, 55.19% of patients were male, and 87.98% of patients were white. There were 21 cases (11.48%) of stage I primary PAAD, 151 cases (82.51%) of stage II PAAD, 3 cases (1.64%) of stage III PAAD, and 5 cases (2.73%) of stage IV PAAD. A total of 53.56% of tumors were histologic grade III, and 77.05% of tumors were located in the head of the pancreas.

## High LACTB mRNA and protein expression in PAAD

PAAD patients and normal patients were downloaded from TCGA and Genotype-Tissue Expression (GTEx) data. The mRNA expression level of LACTB was significantly increased in PAAD tumor tissue (Fig 1A). As shown in Fig 1B, LACTB was able to effectively discriminate between normal pancreatic tissue and pancreatic cancer tissue. In the receiver operating characteristic (ROC) curve analysis of LACTB, the areas under the curve (AUCs) were 0.97 and 0.95 by logistic regression and random forest, respectively, showing that LACTB has high diagnostic value. Moreover, LACTB protein expression has been shown to be regulated at the post-transcriptional level [14,15]. Therefore, 98 primary pancreatic cancer tissues and 69 adjacent noncancerous pancreatic tissues were used to investigate the protein expression level of LACTB by immunohistochemistry (IHC). A score of 1+ to 2+ was defined as low LACTB expression, whereas a score of 3+ was defined as high LACTB expression. As shown in Fig 1C, the proportion of highly expressed LACTB in tumor tissues was significantly higher than that in normal tissues (P = 0.009), and this difference was also found in different sexes (P = 0.004) (Fig 1D). IHC showed that the protein expression of LACTB was strong in tumors (left) and weak in adjacent noncancerous pancreatic tissues (right) (Fig 1E).

## Relationship between LACTB and clinical characteristics of PAAD

To verify the relationship between LACTB expression and clinical characteristics of PAAD patients, the LACTB expression levels of PAAD patients at different clinical stages were analyzed. According to the risk score of OS, all PAAD patients were categorized into LACTB high expression or LACTB low expression groups. As shown in Table 2, the relationship between LACTB mRNA expression and clinical characteristics indicated that LACTB mRNA expression was only significantly associated with vital status and A higher percentage of patients in high LACTB mRNA expression group (61.6%) were decreased compared to the low LACTB mRNA expression group (43.3%) (P = 0.0199). These results suggest that LACTB may be a prognostic factor for PAAD.

## High LACTB mRNA is associated with a poor survival rate

Kaplan-Meier survival curves and log-rank tests were employed to determine the relationship between LACTB and OS. (S1 Fig), DSS (S2 Fig), the progression-free interval (PFI) (S3 Fig) and the disease-free interval (DFI) (S3 Fig). The OS time of 183 patients, the DSS time of 177 patients, the PFI of 183 patients and the DFI of 72 patients were analyzed. The results showed that OS was poor in patients with high LACTB mRNA expression (S1A and S1B Fig; P = 0.039; P = 0.008). Unexpectedly, DSS and PFI analyses produced similar results (S2 and S3 Figs; P = 0.00019 and P = 0.048).

To further find the prognostic value of LACTB in PAAD patients, we conducted OS, DSS, DFI, and PFI analyses in a subgroup of PAAD patients. The subgroup analyses indicated that OS was poor in patients with high LACTB mRNA expression, an age≥60 years, AJCC stage I/II disease, and histological grade G1/G2 disease and in males (S3 Fig), and that DSS was poor in patients with high LACTB mRNA expression, an age ≥ 60 years, AJCC stage I/II disease,

**Table 1. Clinical characteristics of the included patients in TCGA cohort (n = 183).**

| Characteristics | Number of sample (%) |
|---|---|
| **Age(years)** | |
| ≥60 | 125(68.31) |
| <60 | 58(31.69) |
| **Gender** | |
| Male | 101(55.19) |
| Female | 82(44.81) |
| **Histologic grade** | |
| G1 | 31(16.94) |
| G2 | 98(53.56) |
| G3 | 50(27.32) |
| G4 | 2(1.09) |
| Gx | 2(1.09) |
| **Race** | |
| White | 161(87.98) |
| Asian | 12(6.56) |
| Black Or African American | 6(3.28) |
| NA | 4(2.19) |
| **Margin Positiveness** | |
| Yes | 86(46.99) |
| No | 45(24.59) |
| NA | 52(28.42) |
| **Tumor site** | |
| Head | 141(77.05) |
| Body or tail | 31(16.94) |
| Other | 11(6.01) |
| **Tumor stage** | |
| T1 | 7(3.82) |
| T2 | 24(13.11) |
| T3 | 147(80.33) |
| T4 | 3(1.64) |
| Tx | 1(0.55) |
| Na | 1(0.55) |
| **Lymph node status** | |
| N0 | 50(27.32) |
| N1 | 127(69.40) |
| Nx | 5(2.73) |
| Na | 1(0.55) |
| **Metastasis status** | |
| M0 | 81(44.26) |
| M1 | 5(2.73) |
| Mx | 97(53.01) |
| **Ajcc stage** | |
| I | 21(11.48) |
| I I | 151(82.51) |
| III | 3(1.64) |
| IV | 5(2.73) |
| Na | 3(1.64) |

(*Continued*)

**Table 1.** (Continued)

| Characteristics | Number of sample (%) |
|---|---|
| **Vital status** | |
| Living | 88(48.09) |
| deceased | 95(51.91) |

Abbreviations: AJCC = american joint committee on cancer, Na = Not available.

and histological grade G1/G2 disease, male patients, and white patients (S2 Fig); the progression-free interval was poor in patients with high LACTB mRNA expression, AJCC stage I/II disease, and histological grade G1/G2 disease and in males (S3 Fig). Although the DFI showed no significant difference between two LACTB mRNA expression groups (S3 Fig), patients aged ≥60 years showed a poor DFI with high LACTB mRNA expression (S3 Fig). The stratification according tumor location and positive margins of resection were showed in S4 Fig (S4 Fig). OS analysis found that patients with positive margins of resection (P = 0.66), patients with negative margins of resection (P = 0.99) and patients with head of pancreas (P = 0.36) showed no significant difference between two LACTB mRNA expression groups (S1–S3 Figs). However, patients without head of pancreas showed a better OS with high LACTB mRNA expression (P = 0.011) (S4 Fig).

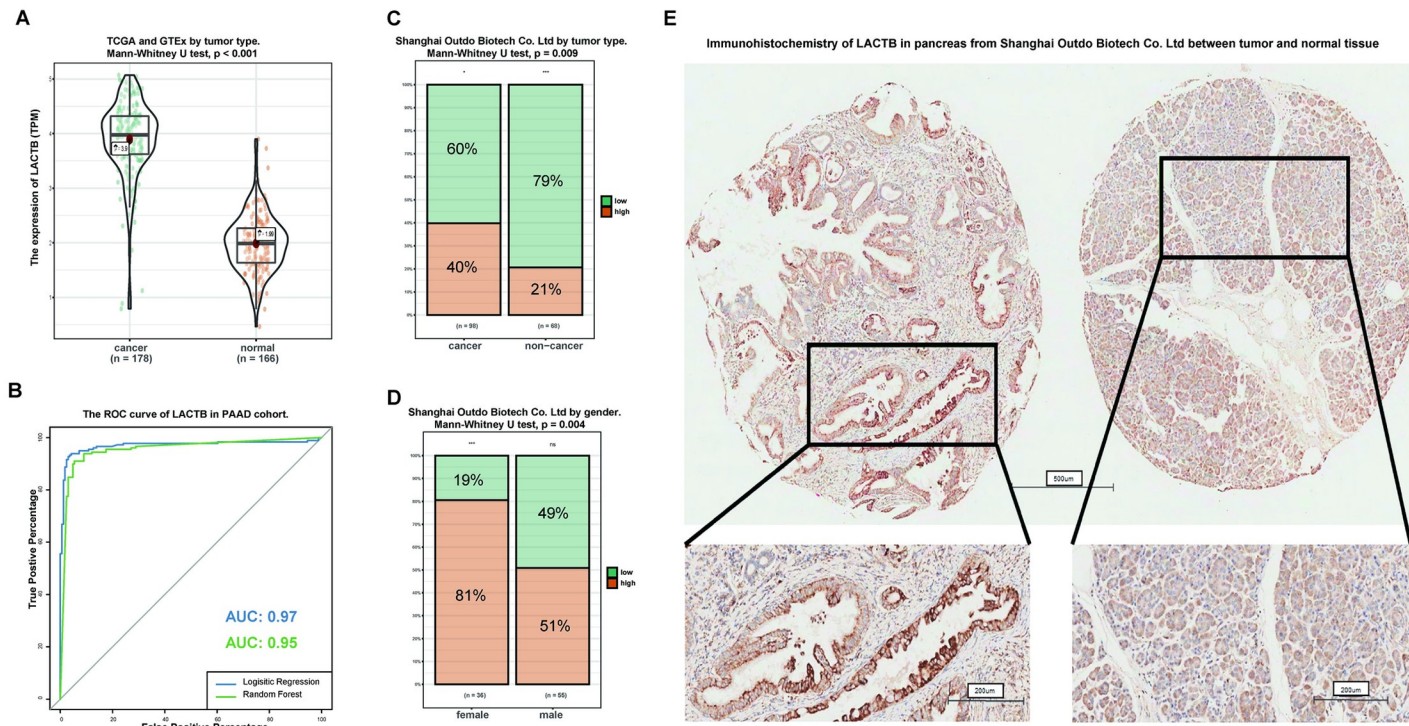

**Fig 1. Validation of the expression and alteration of LACTB in pancreatic cancer.** (A) mRNA expression levels in pancreatic cancer tumor tissue obtained from the TCGA and matching normal tissue obtained from the TCGA and GTEx. Data were obtained from the TCGA and GTEx. (B) The receiver operating characteristic (ROC) curve assessment indicated that LACTB mRNA expression has high diagnostic value in PAAD. (C) Immunohistochemical analysis of the protein expression level of LACTB in pancreatic cancer tumor tissue and normal tissue obtained from Shanghai Outdo Biotech Co., Ltd. (Shanghai, People's Republic of China; Category no: HPan-Ade170Sur-01). (D) Immunohistochemical analysis of the protein expression level of LACTB according to sex. (E) Representative image of the protein expression of LACTB in pancreatic cancer tumor tissue (left) and normal tissue (right).

**Table 2. Correlation between the clinicopathologic variables and LACTB mRNA expression in PAAD.**

| Parameters | Groups | N | LACTB expression | | | | χ2 (P value) |
|---|---|---|---|---|---|---|---|
| | | | High | % | Low | % | |
| Age(years) | ≥60 | 125 | 54 | 62.8 | 71 | 73.2 | 7856.7 (0.177) |
| | <60 | 58 | 32 | 37.2 | 26 | 26.8 | |
| Gender | Female | 82 | 41 | 47.7 | 41 | 42.3 | 174.91 (0.588) |
| | Male | 101 | 45 | 52.3 | 56 | 57.7 | |
| Histologic grade | G1 | 31 | 9 | 10.5 | 22 | 22.7 | 6.9789 (0.137) |
| | G2 | 98 | 51 | 59.3 | 47 | 48.5 | |
| | G3 | 50 | 25 | 29.1 | 25 | 25.8 | |
| | G4 | 2 | 1 | 1.2 | 1 | 1.2 | |
| | GX | 2 | 0 | 0 | 2 | 2.1 | |
| Race | White | 161 | 77 | 89.5 | 84 | 86.6 | 0.645 (0.91) |
| | Nonwhite | 12 | 9 | 10.5 | 13 | 13.4 | |
| Tumor site | Head | 141 | 70 | 81.4 | 71 | 73.2 | 14.505 (0.358) |
| | Body or tail | 31 | 11 | 12.8 | 20 | 20.6 | |
| | Other | 11 | 5 | 5.8 | 6 | 6.2 | |
| AJCC stage | I_II | 172 | 81 | 94.2 | 91 | 93.8 | 12.196 (0.500) |
| | III_IV | 8 | 4 | 4.7 | 4 | 4.7 | |
| Vital status | Living | 88 | 33 | 38.4 | 55 | 56.7 | 5.422 (0.0199*) |
| | Deceased | 95 | 53 | 61.6 | 42 | 43.3 | |

Abbreviations: AJCC = American Joint Committee on Cancer; LACTB = beta-lactamases,

*P<0.05.

## Ability to predict prognosis based on LACTB mRNA expression and protein level

The univariate Cox regression analysis showed that high LACTB mRNA expression was significantly correlated with poor OS (P = 0.001), and other variables, including age (P = 0.001) and histologic grade (P = 0.057), were associated with a reduced OS rate. Multivariate analysis showed that high LACTB mRNA expression (HR = 1.72, P = 0.015, 95% CI = 1.106–2.253) and age (HR = 1.03, P = 0.024, 95% CI = 1.003–1.044) were independent prognostic parameters for the OS of PAAD patients (Fig 2A). In addition, high LACTB mRNA expression (HR = 1.72, P = 0.015, 95% CI = 1.106–2.253) was also an independent prognostic parameter for DSS in patients with PAAD (Fig 2B). In addition, clinico-pathological characteristics of primary tumors included for protein validation was provided in S1 Table. However, we found no predictive value for LACTB at the protein level. The figure was provided in S2 and S3 Tables. All information about the data was supplied in S4 Table.

## Identification of LACTB-related signal transduction pathways by gene set enrichment analysis (GSEA) using the TCGA cohort

We predicted the role of LACTB in PAAD for future research by performing bioinformatic analyses. To confirm the different activation signaling pathways in PAAD, gene expression enrichment analysis was conducted between low and high LACTB mRNA expression datasets. GSEA revealed that HALLMARK_G2M_CHECKPOINT, HALLMARK_MYC_TARGETS_V1, HALLMARK_E2F_TARGETS, and KEGG_CELL_CYCLE were differentially enriched in phenotypes with high LACTB expression (Fig 3A–3D). In addition, CDC20,

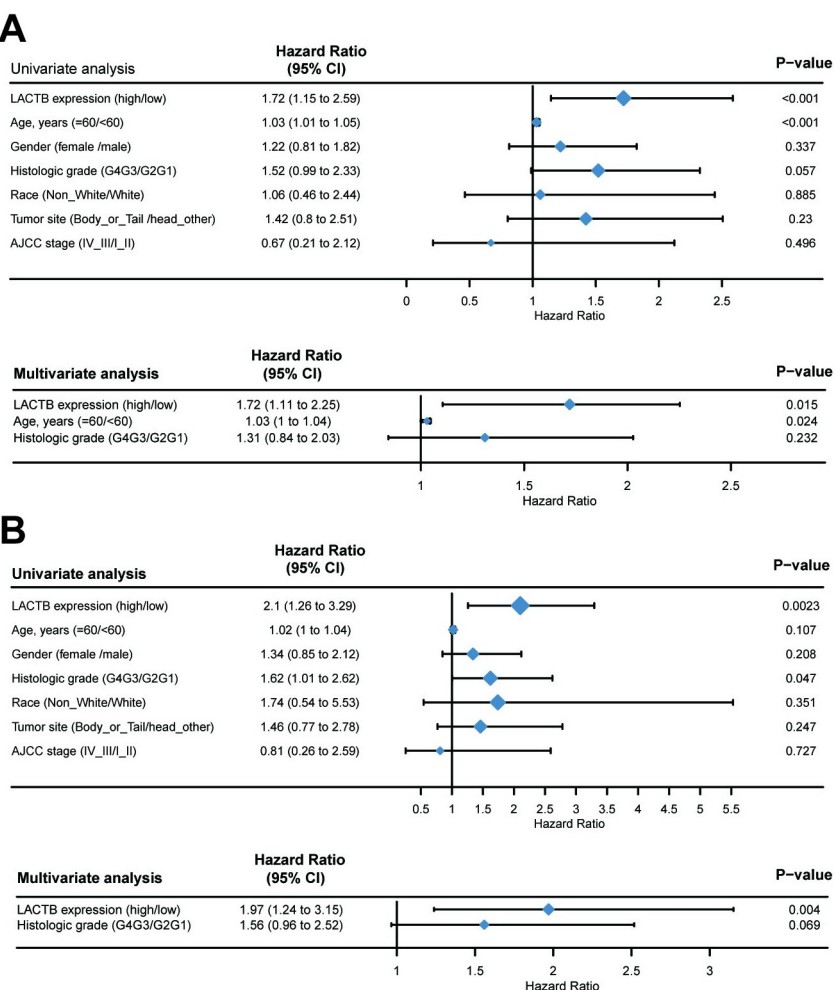

**Fig 2. Univariate and multivariate regression analyses of the relation between the expression of LACTB and clinicopathological characteristics regarding OS (A) and DSS (B) in the TCGA cohort.** We used the COX regression algorithm and p-values less than 0.05 in the univariate analysis were included in the multivariate analysis.

CDK4, MCM6, MAD2L1, MCM2 and MCM5 were leading genes that intersected in these four pathways (Fig 3E). To further investigate the relationships between these genes and LACTB, correlations were analyzed. Our results showed a positive correlation between mRNA expression and LACTB in most normal and cancer tissues (Fig 4A–4L). A PPI network was constructed using the genemania online tool (https://genemania.org/) and showed that these target genes and LACTB exhibited complex interactivity with each other (Fig 5). We regret that we were unable to complete the validation at the protein level and therefore more experiments are needed to validate the protein levels in the future.

## LACTB expression correlates with the landscape of tumor-infiltrating immune cells in PAAD

The ssGSEA function in the gsva R package was used in combination with a signature matrix of 28 immune cell types to calculate the NES of different infiltrating immune cells among patients with different LACTB mRNA expression statuses. Most immune cells correlated with

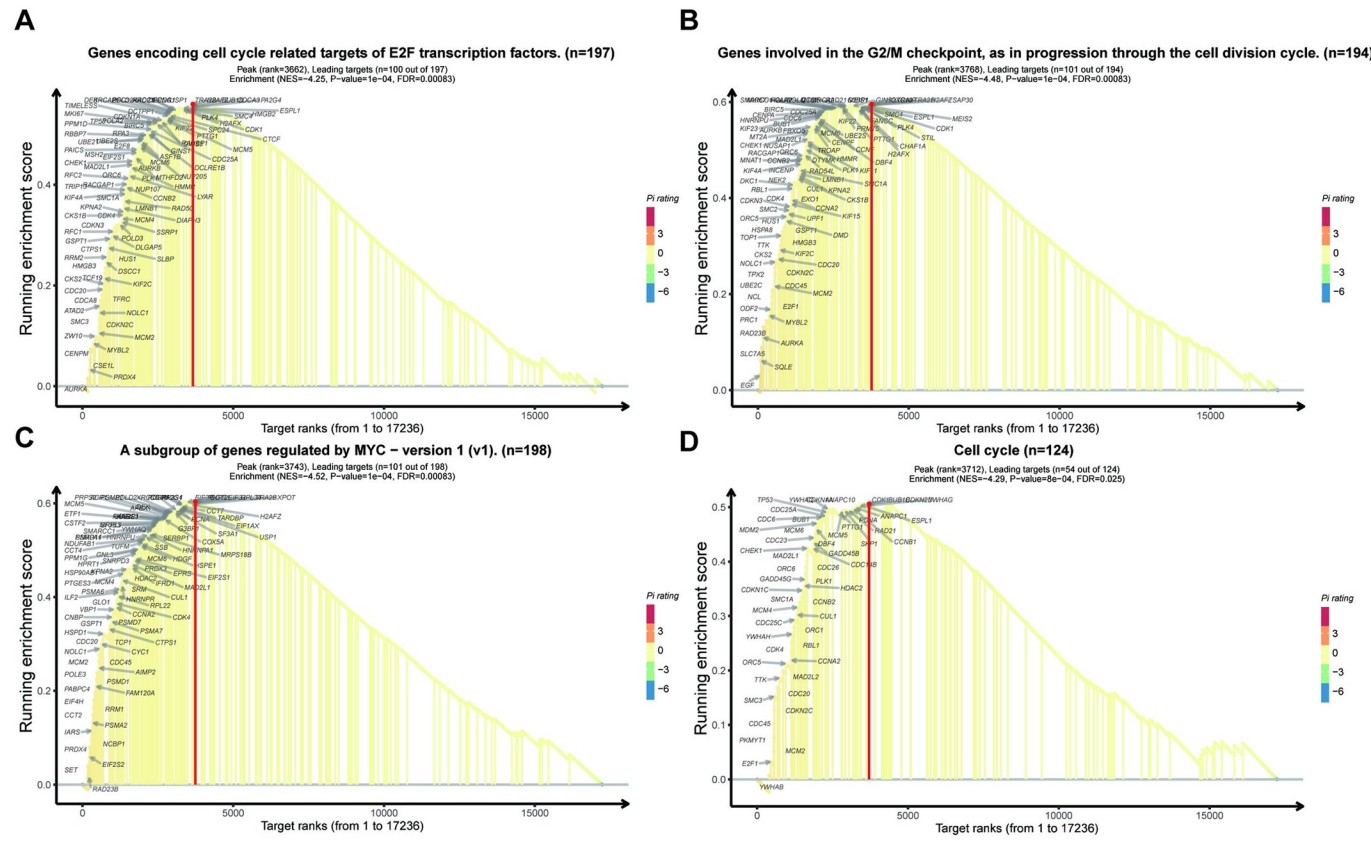

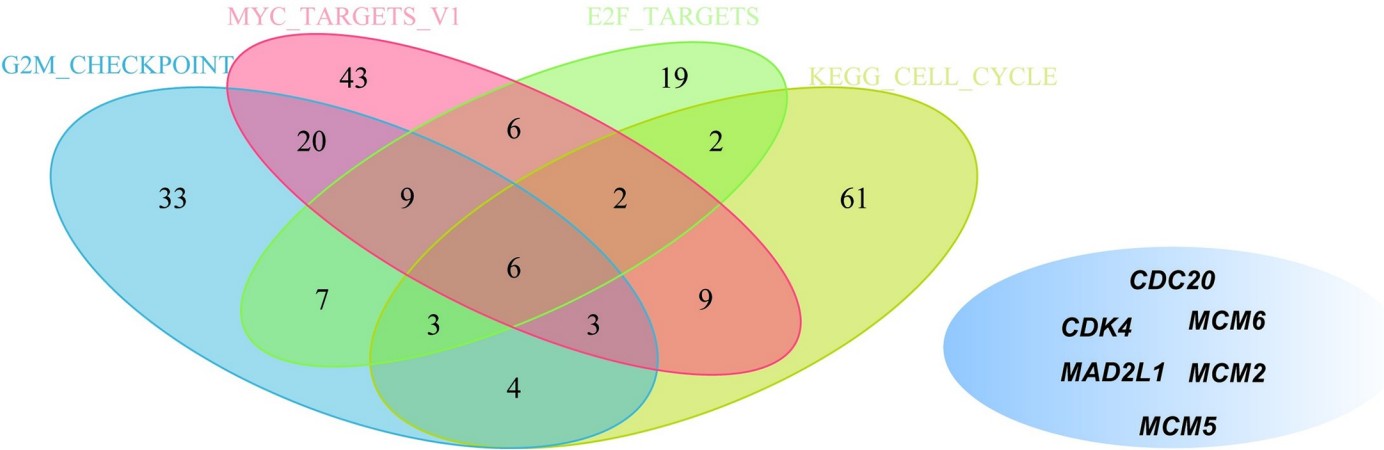

**Fig 3. Enrichment plots from gene set enrichment analysis (GSEA).** GSEA revealed significant differences in the enrichment of (A) hallmark_e2f_targets, (B) hallmark_g2m_checkpoint, (C) hallmark_myc_targets_v1 and (D) kegg_cell_cycle in the TCGA cohort. (E) The leading intersecting genes for these pathways.

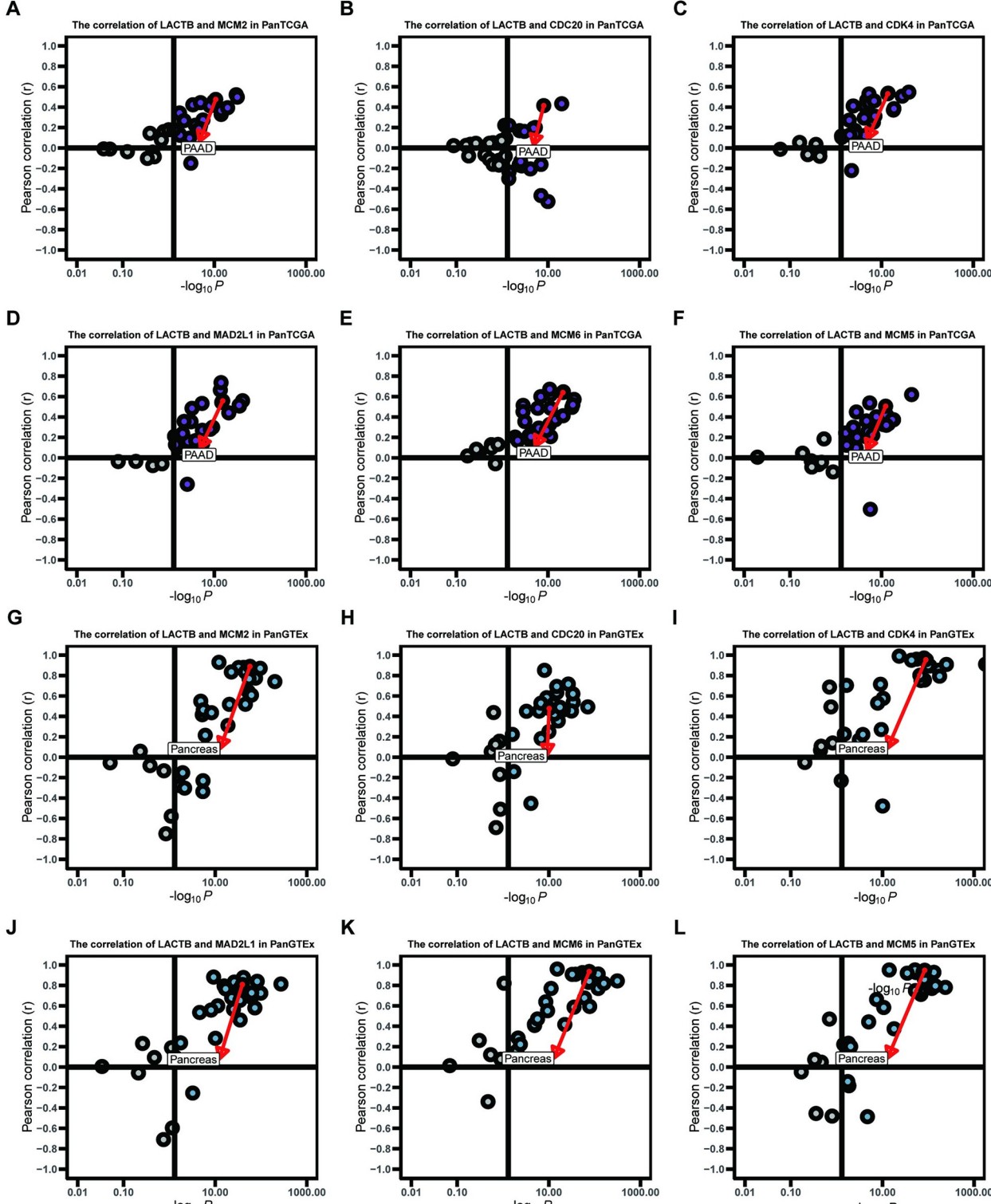

**Fig 4. Correlations of LACTB with the leading intersecting genes (MCM2, CDC20, CDK4, MAD2L1, MCM6 and MCM5) in expression in cancer patients in the TCGA cohort (A-F) and in normal tissues in the GTEx cohort (G-L).**

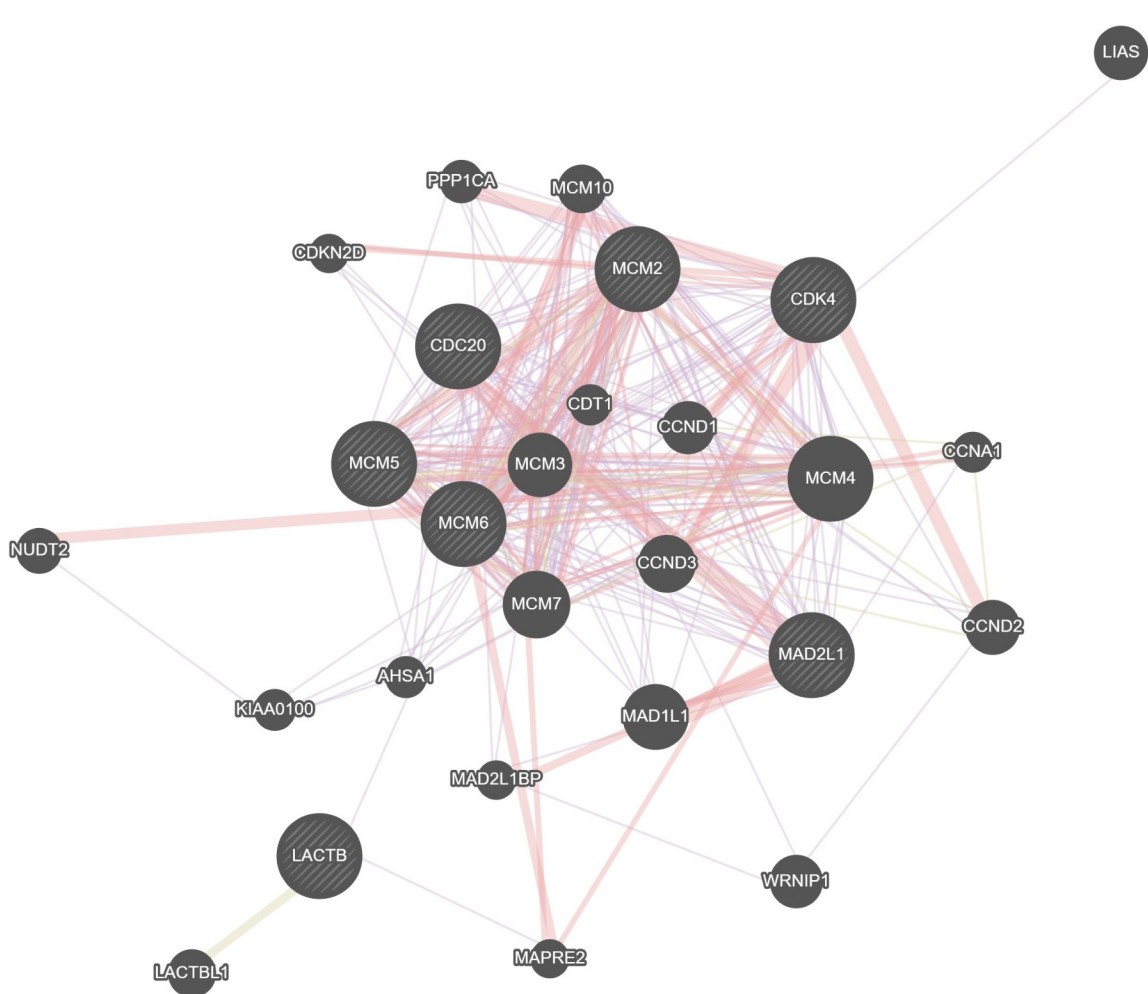

**Fig 5. A PPI network was constructed using the "genemania" online tool.**

LACTB expression (P<0.05) (Fig 6A), and the high LACTB mRNA expression subgroup had a significantly high NES for most immune cell subtypes (Fig 6B). These results indicated that compared with their counterparts, patients in the high LACTB mRNA expression subgroup have a distinct immune phenotype characterized by more immune infiltration.

## Discussion

This study demonstrated the significance of LACTB in PAAD and suggested that LACTB may work as a prognostic biomarker for PAAD. It also showed that high LACTB expression was correlated with the vital status of PAAD patients. Interestingly, although there was no significant difference in the expression of LACTB between AJCC stage I/II and AJCC stage III/IV patients, the expression of LACTB mRNA was higher in deceased than in surviving patients, suggesting that the results might be due to the small sample size of AJCC stage III/IV patients (eight patients); thus, expanding the sample size might provide a more valid result. Because the expression of LACTB mRNA was higher in cancer tissues than in normal tissues in PAAD patients, the relationship between LACTB mRNA and clinical characteristics must be further explored.

**A**

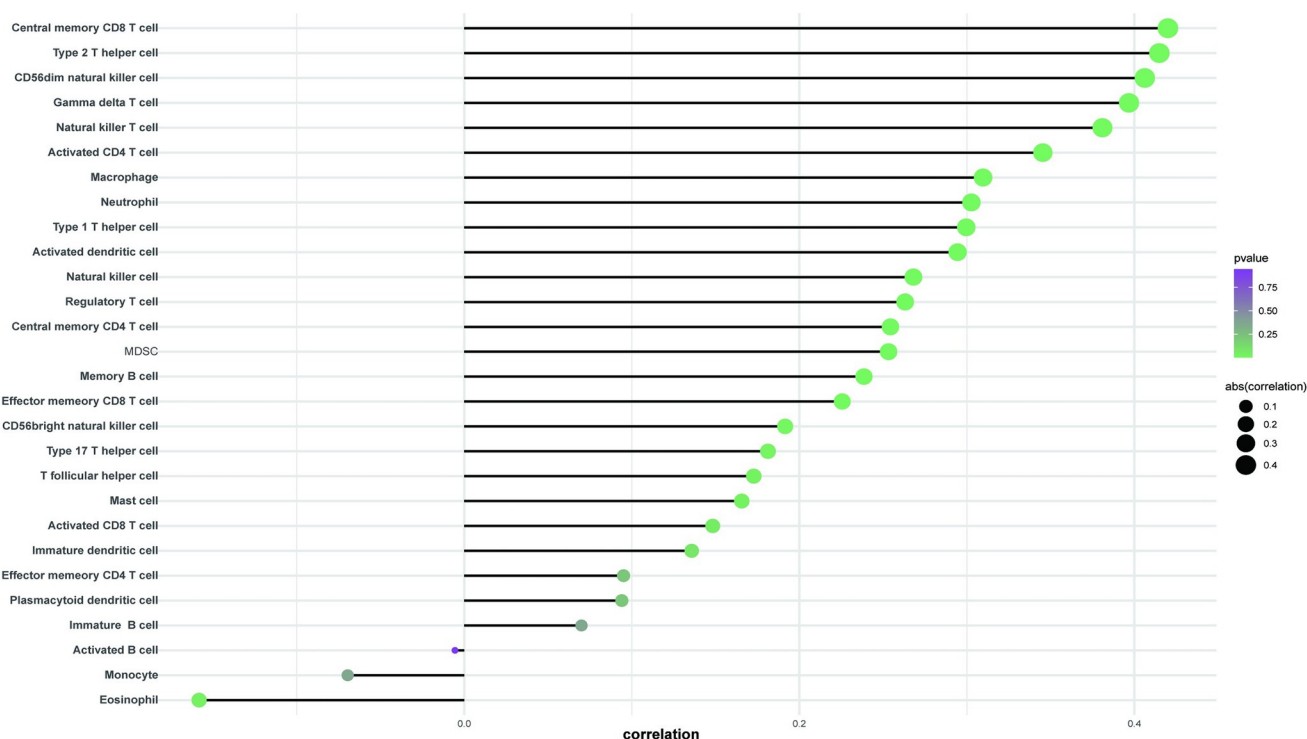

**B**

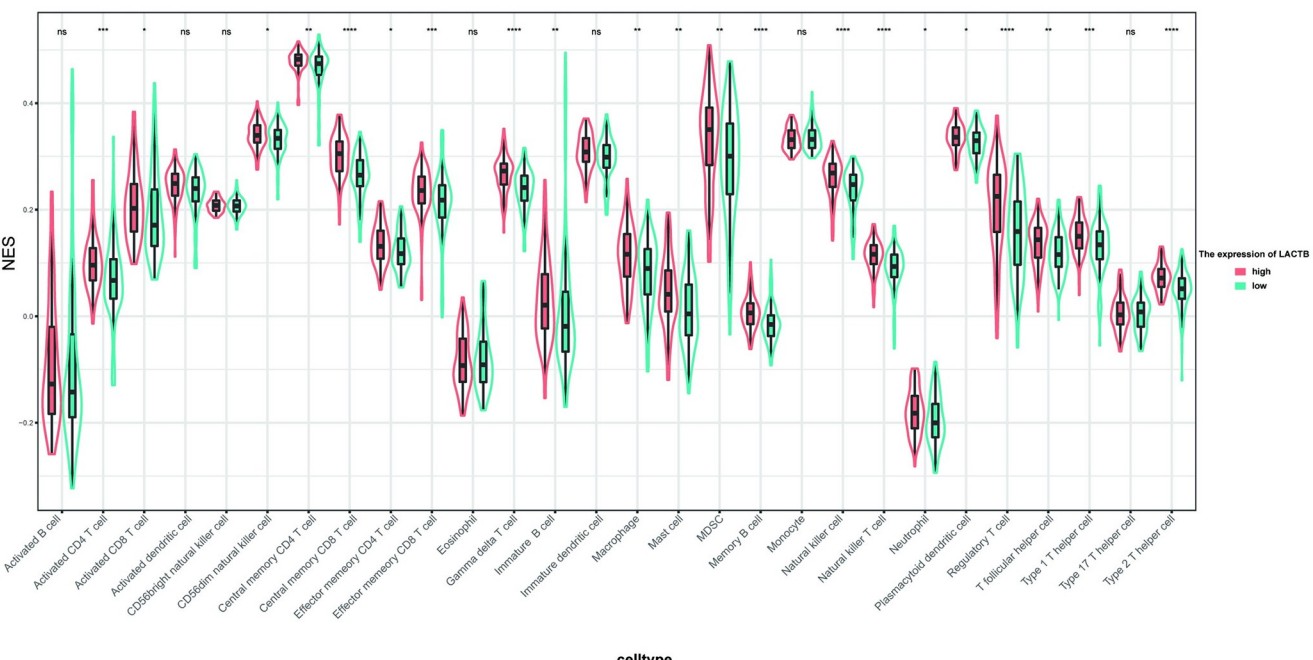

**Fig 6. The ssGSEA function in the gsva R package was used in combination with a signature matrix of 28 immune cell types to calculate the NES of different infiltrating immune cells among patients with different LACTB mRNA expression statuses.** Most immune cells correlated with LACTB expression (P<0.05) (A), and the high LACTB mRNA expression subgroup had a significantly high NES of most immune cell subtypes (B).

LACTB is strongly associated with cancer prognosis. Many studies have used OS as an endpoint, the advantage of which is that it has minimal ambiguity when defining an OS event, and the patient is either alive or dead [16]. However, its drawback is that it may weaken clinical research because deaths from noncancer-related causes do not necessarily reflect tumor biology, invasiveness, or responsiveness to treatment. Therefore, using DSS as an endpoint may improve the accuracy of clinical research, but similar to OS, it also demands longer follow-up times; thus, in many clinical trials, the DFI or PFI is used [16,17]. The shortest follow-up time for these endpoints is shorter because patients usually experience relapse or progression before they die.

It is worth noting that the choice of clinical outcome endpoints for a particular study depends on the objectives of the study, number of events, cohort size, and quality of the outcome data [18]. Jianfang Liu et al [18] provided recommendations regarding how each outcome's endpoints should be used within each disease type, with concerns justified in comments, and they recommend the use of all four endpoints without reservation for 13 of the 33 cancer types including PAAD in the TCGA database.

Therefore, in our study, all four endpoints were analyzed, However, in the subgroup analysis of AJCC stage I/II, histological grade G1/G2 and male sex were associated with a poor PFI, and in the subgroup analysis of patients ≥60 years old, the DFI was poor in those with high LACTB mRNA expression. However, inconsistent with our research, low LACTB is closely related to the poor prognosis of many cancers. Hai-Tao Li et al. demonstrated that the downregulation of LACTB was significantly related to poor OS in glioma cells and that LACTB expression was a prognostic factor for gliomas [8]. Kaixuan Zeng et al. also showed that low LACTB expression was an independent prognostic factor for the poor OS of colorectal cancer patients [9]. Chen Xue et al [10] found that LACTB mRNA was downregulated and contributed to the unfavorable prognosis of hepatocellular carcinoma patients. In addition, we found patients with positive margins of resection, patients with negative margins of resection and patients with head of pancreas showed no significant OS difference between two LACTB mRNA expression groups in the subgroup survival analysis, however, patients without head of pancreas showed a better OS with high LACTB mRNA expression. Therefore, our analysis of the physiological function of LACTB may also differ in different subgroups and need further analysis. This discrepancy suggests that the real roles of LACTB vary in different cancer types and that other unreported mechanisms may be involved in the effects of LACTB in PAAD.

LACTB is a mitochondrial protein that is related to the evolution of bacterial penicillin-binding/B-lactamase proteins [5]. LACTB has been shown to be widespread in different mammalian tissues, most notably in the skeletal muscle, heart and liver [6]. Several studies have demonstrated that LACTB is strongly related to high-density lipoprotein cholesterol [19,20], and Bains Randip K et al. showed that LACTB deletion leads to late-onset obesity in transgenic mice [21]. However, there are few studies on the role of LACTB in tumorigenesis and progression. Until recently, Keckesova et al. showed that LACTB was a tumor suppressor and inhibited the proliferation and promoted the apoptosis of breast cancer cells by inhibiting the proliferation of many types of breast cancer cells, and LACTB has the ability to change mitochondrial lipid metabolism and regulate the differentiation of cancer cells through this reprogramming, which is achieved through the LACTB-PISD-LPE/PE signaling axis. Subsequently, the tumor-suppressive effects and prognostic values of LACTB were demonstrated in many different cancer types. Hai-Tao Li et al [8]. showed that the downregulated expression of LACTB is correlated with a poor prognosis of glioma and revealed that LACTB is an independent prognostic indicator for glioma patients. In addition, the overexpression of LACTB can inhibit the expression of PCNA, MMP2, MMP9 and VEGF [8], which are believed to play an important role in the proliferation, invasion and angiogenesis of glioma cells [22]. Kaixuan

Zeng et al. found that low LACTB expression was associated with poor OS in colorectal cancer patients, and LACTB was also determined to be an independent prognostic factor for poor outcomes. The mechanism involves LACTB binding directly to the C terminus of p53 and inhibiting the degradation of p53 by preventing the interaction between MDM2 and p53. In addition, the ablation of p53 attenuates the antitumorigenic effects of LACTB overexpression in colorectal cancer [9]. Additionally, Chen Xue et al. demonstrated that both LACTB mRNA and protein levels are downregulated in hepatocellular carcinoma and that low LACTB expression is associated with poor OS and relapse-free survival. An online prediction suggested that the LACTB gene is markedly correlated with genes involved in the lipid metabolism pathway [10]. However, in this study, the LACTB gene was recognized as an oncogene, and the mRNA expression and protein expression of LACTB were significantly higher in PAAD tumor tissue than in adjacent normal tissue. Moreover, the high expression of LACTB mRNA was an independent prognostic factor for OS and DSS in patients with PAAD.

Moreover, GSEA further predicted the potential role of LACTB in PAAD. The seven most significantly enriched signal transduction pathways were found. According to research, some of them, including the hallmark early estrogen response pathway, have been identified as cancer-promoting pathways.

High LACTB expression was positively associated with hallmark_g2m_checkpoint, hallmark_myc_targets_v1, hallmark_e2f_targets and kegg_cell_cycle. In addition, Cell division cycle 20 (CDC20), Cyclin-dependent kinase 4 (CDK4), minichromosome maintenance 6 (MCM6), MAD2L1, minichromosome maintenance 2 (MCM2) and minichromosome maintenance 5 (MCM5) were leading genes that intersected in these four pathways, and their mRNA expression levels were positively correlated with LACTB in most normal and cancer tissues. CDC20 is an activator of the division-promoting complex necessary for cell division. David Z Chang et al [23] suggested that CDC20 may play a key role in the development and progression of pancreatic cancer and thus may serve as a marker of disease progression and prognosis as well as a therapeutic target. CDK4 is a key regulator of the G1 phase of the cell cycle and has been shown to be overexpressed in pancreatic cancer. Michaela Retzer-Lidl et al. [24] found that inhibition of CDK4 impairs the proliferation of pancreatic cancer cells and sensitizes them to TRAIL-induced apoptosis. MCM genes including MCM2, MCM5 and MCM6 were upregulated in pancreatic cancer and show strong positive coexistence with each other [25]. Finally, elevated LACTB mRNA expression was significantly related to multiple immune marker sets.

## Conclusions

Our results suggest that LACTB is involved in cancer progression and that high LACTB expression in PAAD patients predicts poor prognosis. High LACTB expression is significantly associated with cell cycle-related genes and multiple immune marker sets.

## Supporting information

**S1 Fig. Survival analysis of LACTB expression in terms of overall survival (OS).** OS values were analyzed in relation to the mRNA expression level of LACTB in all tumors and subgroups of PAAD patients. OS analyses of (A) all tumors (divided according to the median LACTB expression levels), (B) all tumors (divided according to the best separation), (C) G1 +G2 stage, (D) age $\geq$ 60 years, (E) AJCC stage I/II and (F) male sex.
(TIF)

**S2 Fig. Survival analysis of LACTB expression in terms of disease-specific survival (DSS).**
DSS values were analyzed in relation to the mRNA expression level of LACTB in all tumors
and subgroups of PAAD patients. DSS analyses of (A) all tumors, (B) AJCC stage I/II, (C) male
sex, (D) age $\geq$ 60 years, (E) G1 +G2 stage and (F) white race.
(TIF)

**S3 Fig. Survival analysis of LACTB expression in terms of the progression-free interval
(PFI) and disease-free interval (DFI).** PFI and DFI values were analyzed in relation to the
mRNA expression level of LACTB in all tumors and subgroups of PAAD patients. PFI analysis
of (A) all tumors, (B) AJCC stage I/II, (C) G1 +G2 stage, and (D) male sex; DFI analysis of (E)
all tumors and (F) age > 60 years.
(TIF)

**S4 Fig. Survival analysis of LACTB expression in terms of the overall survival (OS).** OS
analyses of (A) positive margins of resection (B) negative margins of resection, (C) tumor loca-
tion with head of pancreas (D) tumor location without head of pancreas.
(TIF)

**S1 Table.**
(DOCX)

**S2 Table.**
(CSV)

**S3 Table.**
(CSV)

**S4 Table.**
(CSV)

## Author Contributions

**Conceptualization:** Jian Xie, Yang Peng, Qigang Li.

**Data curation:** Bin Jian, Shengchun Liu.

**Project administration:** Xiaoyu Chen, Qigang Li, Zelin Wen.

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
