## [Decision Letter · Decision Letter 0]

23 Oct 2020

PONE-D-20-30765

LACTB mRNA expression is increased in pancreatic adenocarcinoma and indicates a poor prognosis

PLOS ONE

Dear Dr. Liu,

Thank you for submitting your manuscript to PLOS ONE. After careful consideration, we feel that it has merit but does not fully meet PLOS ONE’s publication criteria as it currently stands. Therefore, we invite you to submit a revised version of the manuscript that addresses the points raised during the review process.

This manuscript was carefully reviewed by 2 experts, and both of them found a number of concerns which need to be addressed before acceptance. For instance, reviewer 1 suggested additional analyses to further consolidate the results. Reviewer 2 raised several questions regarding the analysis results. Please respond to each of the reviewer comments.

We look forward to receiving your revised manuscript.

Kind regards,

Hiromu Suzuki, M.D., Ph.D.

Academic Editor

PLOS ONE

Journal Requirements:

2. Please provide the accession numbers and/or URLs of the datasets obtained from TCGA and GTEx.

3. We note that you obtained a tissue microarray from Shanghai Outdo Biotech Co. Please ensure it is clear to readers that your study did not involve prospective collection of tissue samples. Specifically, please ensure that you do not refer to "patients recruitment".

4. In your ethics statement in the manuscript and in the online submission form, please provide additional information about the patient data used in your study. Specifically, please ensure that you have discussed whether all data were fully anonymized before you accessed them.

5. To comply with PLOS ONE submission guidelines, in your Methods section, please provide additional information regarding your statistical analyses. For more information on PLOS ONE's expectations for statistical reporting, please see https://journals.plos.org/plosone/s/submission-guidelines.#loc-statistical-reporting.

Reviewers' comments:

Reviewer's Responses to Questions

**Comments to the Author**

1. Is the manuscript technically sound, and do the data support the conclusions?

Reviewer #1: No

Reviewer #2: Yes

2. Has the statistical analysis been performed appropriately and rigorously? 

Reviewer #1: No

Reviewer #2: No

3. Have the authors made all data underlying the findings in their manuscript fully available?

Reviewer #1: No

Reviewer #2: Yes

4. Is the manuscript presented in an intelligible fashion and written in standard English?

Reviewer #1: Yes

Reviewer #2: Yes

5. Review Comments to the Author

Reviewer #1: The article entitled: LACTB mRNA expression is increased in pancreatic adenocarcinoma and indicates a poor prognosis by Jian Xie et al. is an interesting study about the clinical impact of LACTB expression levels. The manuscript is overall well written and introduced; however, the study presents several flaws and lacks that make conclusions unsupported by results:

Minors

-Title is ambiguous, please specify if high or low levels are associated to poor prognosis.

-Include an updated information about 5-years survival rates, mortality, presence of metastasis, accurate prognosis biomarkers, etc. Please try to provide an exact number in each case and avoid generalities.

-In Table 1 headline please refer to TCGA cohort. In addition, provide an accurate stratification of patient by ethnicity, white and non-white is a rough estimation. Please include margin positiveness in case of resected patients, and include this variable in analyses. And include in abbreviations the meanings of NA, G, T, N and M.

-Explain the cut-off point when mRNA expression levels are used.

- In "High LACTB mRNA is associated with a poor survival rate" section OS, DSS, DFI, and PFI analyses was assessed in a subgroup of PAAD patients. Please explain inclusion criteria of this subgroup.

-Please check the order of the variables included in the uni- and multi-variate analyses

; e.g. HR of high vs Low expression LACTB is 1.72 and HR of stage IV/III vs I/II is 0.67 what does not make any sense.

-Figure legend of Fig.2 is not well described.

Majors

-The study has been carried out with a high heterogeneous TCGA cohort that includes resectable and non-resectable tumors and several tumor locations. Please re-analyse with selected patients or do a stratification according to stage, tumor location and positive margins of resection.

- Include clinico-pathological characteristics of 98 primary tumors included for protein validation.

-Include a uni- and multi-variate analysis for survival of the primary tumors included for protein validation.

-Since LACTB presents a clear cytoplasmic staining, score has been performed with both intensity of expression and % of positive staining cells. Please justify why just intensity has been taken into consideration.

-Association/correlation with leading genes intersecting with LACTB must be validated at least at protein expression.

Reviewer #2: The Authors have done some considerable work to show that up-regulated LACTB expression can be used as a predictive marker for PAAD. There are however some major concerns regarding some of the analysis work that will need to be clarified or modified:

1. Figures were not visible for review and that hindered the ability to review any Fig from the paper. Please paste the figures within the submitted draft so that it can be reviewed

2. Under the section ImmunoHistochemical Staining it says: "For IHC analysis, a tissue microarray including 98 primary pancreatic cancer tissues and 68 adjacent noncancerous pancreatic tissues...". It is not clear from the statement whether the cancerous and non cancerous samples were taken from the same patient/different patient? Are the two sample datasets mutually exclusive vis-a-vis PAAD? If they were from the same set of individuals, is there a reason for only 68 noncancerous tissue samples versus 98 cancerous.

3. Also how is the PAAD patient table broken down by age/ethnicity etc..

4. Table 2 provides a breakdown of the LACTB expression but the % values will need to be explained. I could not figure out how the numbers 62.8% and 73.2 were calculated.

5. Here: "The chi-square test was performed to examine the relationship between LACTB mRNA expression and clinical data"

This section, what "clinical data" are we comparing to? there is no reference. The Chi Square test will need to be clarified better if the samples aren't assumed to be mutually exclusive..

6. PLOS authors have the option to publish the peer review history of their article (what does this mean?). If published, this will include your full peer review and any attached files.

Reviewer #1: No

Reviewer #2: No

---

## [Author Response · Author response to Decision Letter 0]

18 Nov 2020

Journal Requirements:

The article format has been modified as required 

2. Please provide the accession numbers and/or URLs of the datasets obtained from TCGA and GTEx.

All URLs a were supplied on data availability.

3. We note that you obtained a tissue microarray from Shanghai Outdo Biotech Co. Please ensure it is clear to readers that your study did not involve prospective collection of tissue samples. Specifically, please ensure that you do not refer to "patients recruitment".

We have modified the statement accordingly to ensure that the reader understands correctly, we have changed “patients recruitment” to “samples collected”.

4. In your ethics statement in the manuscript and in the online submission form, please provide additional information about the patient data used in your study. Specifically, please ensure that you have discussed whether all data were fully anonymized before you accessed them.

All additional information on the use of patients has been submitted as additional information and full anonymity is ensured.

5. To comply with PLOS ONE submission guidelines, in your Methods section, please provide additional information regarding your statistical analyses. For more information on PLOS ONE's expectations for statistical reporting, please see https://journals.plos.org/plosone/s/submission-guidelines.#loc-statistical-reporting.

To facilitate reproduction of article results, all analysis data have been provided as additional information.

Reviewer #1:

Minors :

(1) Title is ambiguous, please specify if high or low levels are associated to poor prognosis.

To eliminate ambiguous, we changed the title to “LACTB mRNA expression is increased in pancreatic adenocarcinoma and high expression indicates a poor prognosis”

(2) Include an updated information about 5-years survival rates, mortality, presence of metastasis, accurate prognosis biomarkers, etc. Please try to provide an exact number in each case and avoid generalities.

All the above-mentioned issues have been corrected

(3) In Table 1 headline please refer to TCGA cohort. In addition, provide an accurate stratification of patient by ethnicity, white and non-white is a rough estimation. Please include margin positiveness in case of resected patients and include this variable in analyses. And include in abbreviations the meanings of NA, G, T, N and M.

All the above-mentioned issues have been corrected, eg: The stratification of patient by ethnicity has been updated. T = Tumor stage, N = Lymph node status, M = Metastasis status, G = Histologic grade, NA = Not available. 

(4) Explain the cut-off point when mRNA expression levels are used.

The samples were divided into high-risk and low-risk groups based on the median LACTB expression levels.

(5) In "High LACTB mRNA is associated with a poor survival rate" section OS, DSS, DFI, and PFI analyses was assessed in a subgroup of PAAD patients. Please explain inclusion criteria of this subgroup.

To ensure more accurate results, we expected to include as many patients as possible in the analysis; however, not all patients in the TCGA database had OS, DSS, DFI, and PFI information collected, so we only analyzed patients for whom OS (183), DSS (177 patients), DFI (72 patients), and PFI (183 patients) information existed.

(6) Please check the order of the variables included in the uni- and multi-variate analyses e.g. HR of high vs Low expression LACTB is 1.72 and HR of stage IV/III vs I/II is 0.67 what does not make any sense.

We repeated the calculations, but came to the same conclusion, suggesting that the results might be due to the small sample size of AJCC stage III/IV patients (eight patients); thus, expanding the sample size might provide a more valid result.

(7) Figure legend of Fig.2 is not well described

Fig 2. Univariate and multivariate regression analyses of the relation between the expression of LACTB and clinicopathological characteristics regarding OS (A) and DSS (B) in the TCGA cohort. We used the COX regression algorithm and p-values less than 0.05 in the univariate analysis were included in the multifactor analysis

Majors

1. The study has been carried out with a high heterogeneous TCGA cohort that includes resectable and non-resectable tumors and several tumor locations. Please re-analyse with selected patients or do a stratification according to stage, tumor location and positive margins of resection.

The stratification according to stage, tumor location and positive margins of resection were provided in supplement figures. 

2. Include clinico-pathological characteristics of 98 primary tumors included for protein validation.

The data was provided in supplement Table 1.

3. Include a uni- and multi-variate analysis for survival of the primary tumors included for protein validation.

We found no predictive value for LACTB at the protein level. The figure was provided in supplement Table 2 and supplement Table 3. All information about the data was supplied in supplement Table 4.

4. Since LACTB presents a clear cytoplasmic staining, score has been performed with both intensity of expression and % of positive staining cells. Please justify why just intensity has been taken into consideration.

To facilitate subsequent statistical analysis, LACTB was scored according to staining intensity from 1+ to 3+. A score of 1+ to 2+ was defined as low LACTB expression, whereas a score of 3+ was defined as high LACTB expression.

5. Association/correlation with leading genes intersecting with LACTB must be validated at least at protein expression.

CDC20, CDK4, MCM6, MAD2L1, MCM2 and MCM5 were leading genes that intersected in signal transduction pathways by gene set enrichment analysis (GSEA). There is a strong correlation between these genes at the molecular level and LACTB. A PPI network was constructed using the genemania online tool(https://genemania.org/) and showed that these target genes and LACTB exhibited complex interactivity with each other (Fig 5). Therefore, we predict that LACTB may function biologically together with these genes. However, more experiments are needed to validate the protein levels. 

Reviewer #2: 

1. Figures were not visible for review and that hindered the ability to review any Fig from the paper. Please paste the figures within the submitted draft so that it can be reviewed

All figures have been provided.

2. Under the section ImmunoHistochemical Staining it says: "For IHC analysis, a tissue microarray including 98 primary pancreatic cancer tissues and 68 adjacent noncancerous pancreatic tissues...". It is not clear from the statement whether the cancerous and non cancerous samples were taken from the same patient/different patient? Are the two sample datasets mutually exclusive vis-a-vis PAAD? If they were from the same set of individuals, is there a reason for only 68 noncancerous tissue samples versus 98 cancerous.

We apologize for using such a puzzling description. We have made the following corrections: For IHC analysis, a tissue microarray including 98 primary pancreatic cancer tissues and 68 noncancerous pancreatic tissues was obtained from Shanghai Outdo Biotech Co., Ltd. (Shanghai, People’s Republic of China; Category no: HPan-Ade170Sur-01).

3. Also how is the PAAD patient table broken down by age/ethnicity etc..

To facilitate statistical analysis, we set the age to 60 as the cutoff point and White and non-white as ethnicity.

4. Table 2 provides a breakdown of the LACTB expression but the % values will need to be explained. I could not figure out how the numbers 62.8% and 73.2 were calculated.

That's for the chi-square test, represents in high LACTB expression patients 62.8% patients more than 60 years old and 37.2% patients less than 60 years old; In addition, in low LACTB expression patients, 73.2% patients more than 60 years old 26.8% patients less than 60 years old.

5. Here: "The chi-square test was performed to examine the relationship between LACTB mRNA expression and clinical data"

This section, what "clinical data" are we comparing to? there is no reference. The Chi Square test will need to be clarified better if the samples aren't assumed to be mutually exclusive..

To prevent misunderstandings, we have modified the statement as follows: The chi-square test was performed to examine the clinical relationship between high and low LACTB mRNA expression patients

---

## [Decision Letter · Decision Letter 1]

24 Dec 2020

PONE-D-20-30765R1

LACTB mRNA expression is increased in pancreatic adenocarcinoma and high expression indicates a poor prognosis

PLOS ONE

Dear Dr. Liu,

Thank you for submitting your manuscript to PLOS ONE. After careful consideration, we feel that it has merit but does not fully meet PLOS ONE’s publication criteria as it currently stands. Therefore, we invite you to submit a revised version of the manuscript that addresses the points raised during the review process.

The authors addressed many of the issues raised by the reviewers. However, reviewers indicated several revisions to improve the manuscript. Please respond to each of the reviewer comments.

We look forward to receiving your revised manuscript.

Kind regards,

Hiromu Suzuki, M.D., Ph.D.

Academic Editor

PLOS ONE

Reviewers' comments:

Reviewer's Responses to Questions

**Comments to the Author**

1. If the authors have adequately addressed your comments raised in a previous round of review and you feel that this manuscript is now acceptable for publication, you may indicate that here to bypass the “Comments to the Author” section, enter your conflict of interest statement in the “Confidential to Editor” section, and submit your "Accept" recommendation.

Reviewer #1: (No Response)

Reviewer #2: All comments have been addressed

2. Is the manuscript technically sound, and do the data support the conclusions?

Reviewer #1: Partly

Reviewer #2: Partly

3. Has the statistical analysis been performed appropriately and rigorously? 

Reviewer #1: No

Reviewer #2: Yes

4. Have the authors made all data underlying the findings in their manuscript fully available?

Reviewer #1: Yes

Reviewer #2: Yes

5. Is the manuscript presented in an intelligible fashion and written in standard English?

Reviewer #1: Yes

Reviewer #2: Yes

6. Review Comments to the Author

Reviewer #1: 1.-Please describe in "Results" section those findings after survival analyses according to tumor location and positive margins of resection and discuss them in "Discussion" section.

2.-Please include a survival analysis of R0 patients stratified by LACTB expression (mRNA and protein), and in R1 patients stratified by LACTB expression (mRNA and protein).

3.-Previous major point 5 has not been amended. As least, include in the "Results" section that more experiments are needed to validate the protein levels

4.-Include in Table S2 whether patients are R0 or R1

5.-Include in Table S4 the R0 or R1 status of each patient.

Reviewer #2: The authors have taken the time to address a lot of the questions posed. Many of the clarifications they provided also help better the understanding of their analyses. However, there are a few concerns which, if clarified, will greatly enhance the understanding of the data and increase appreciation for the analyses conducted.

They are listed herewith:

1. your Statistical analysis lists a few metrics: OS, DSS, DFI, PFI, NES etc.. these need to be elucidated. Many of them are not expanded until later in the Supplementary. It will help if these terms are explained when they are first introduced in the manuscript to avoid any confusion.

2. The use of 'Samples' and 'patients' is constantly switched back and forth. It would be preferable if the authors consistently used samples throughout the paper.

3. Table 1: change Number of sample size to either "Number of samples" or "Sample size" or "percentage of samples"

4. Table 2: Please elucidate further on table2. You show how LACTB expression shows a significant association with Vital status, but it would help if you could further elucidate on what the implications are.. or drive conclusions on what the p-value shows in this table.

5. it would help strengthen the paper if you could provide a power analysis: that shows the power of your test based on your sample size.

7. PLOS authors have the option to publish the peer review history of their article (what does this mean?). If published, this will include your full peer review and any attached files.

Reviewer #1: No

Reviewer #2: No

---

## [Author Response · Author response to Decision Letter 1]

24 Dec 2020

We thank the reviewers for their valuable comments, all of questions have been carefully reviewed and revised.

Reviewer #1:

1.-Please describe in "Results" section those findings after survival analyses according to tumor location and positive margins of resection and discuss them in "Discussion" section.

We have added the section as follows: 

Results: OS analysis found that patients with positive margins of resection (P = 0.66), patients with negative margins of resection (P = 0.99) and patients with head of pancreas (P = 0.36) showed no significant difference between two LACTB mRNA expression groups (S4 Fig.1-3). However, patients without head of pancreas showed a better OS with high LACTB mRNA expression (P=0.011) (S4 Fig.4).

Discussion: However, patients without head of pancreas showed a better OS with high LACTB mRNA expression. Therefore, our analysis of the physiological function of LACTB may also differ in different subgroups and need further analysis. This discrepancy suggests that the real roles of LACTB vary in different cancer types and that other unreported mechanisms may be involved in the effects of LACTB in PAAD.

2.-Please include a survival analysis of R0 patients stratified by LACTB expression (mRNA and protein), and in R1 patients stratified by LACTB expression (mRNA and protein).

We interpret R0 as patients with negative margins of resection, and R1 as patients with positive margins of resection. However, the protein data of LACTB expression were obtained from Shanghai Outdo Biotech Co., Ltd. (Shanghai, People’s Republic of China; Category no: HPan-Ade170Sur-01), they did not record the corresponding information about margins of tumor resection. Therefore, we present the results of OS analyses of positive margins of resection and negative margins of resection between high and low LACTB mRNA expression in supplement figure 4. And added the corresponding text description in the Results and Discussion section. 

3.-Previous major point 5 has not been amended. As least, include in the "Results" section that more experiments are needed to validate the protein levels

We regret that we were unable to complete the validation at the protein level and therefore we note in the results section as follows: more experiments are needed to validate the protein levels in the future.

4.-Include in Table S2 whether patients are R0 or R1

5.-Include in Table S4 the R0 or R1 status of each patient.

Here we make the following explanation: Table S2 and Table S4 showed the clinico-pathological characteristics and predictive value of LACTB in protein level in PAAD. However, the protein data of LACTB expression were obtained from Shanghai Outdo Biotech Co., Ltd. (Shanghai, People’s Republic of China; Category no: HPan-Ade170Sur-01), they did not record the corresponding information about R0 and R1. Therefore, we cannot add the above information in Table S2 and Table S4. However, the survival analysis of R0/R1 patients stratified by LACTB mRNA expression were supplied in supplement figure 4. 

Reviewer #2: 

1.- your Statistical analysis lists a few metrics: OS, DSS, DFI, PFI, NES etc.. these need to be elucidated. Many of them are not expanded until later in the Supplementary. It will help if these terms are explained when they are first introduced in the manuscript to avoid any confusion.

We have annotated all abbreviations where they first appear

2. The use of 'Samples' and 'patients' is constantly switched back and forth. It would be preferable if the authors consistently used samples throughout the paper.

We have changed all the 'Samples' in the original text to 'patients'

3. Table 1: change Number of sample size to either "Number of samples" or "Sample size" or "percentage of samples"

The issues mentioned above have been modified

4. Table 2: Please elucidate further on table2. You show how LACTB expression shows a significant association with Vital status, but it would help if you could further elucidate on what the implications are.. or drive conclusions on what the p-value shows in this table.

A higher percentage of patients in high LACTB mRNA expression group (61.6%) were decreased compared to the low LACTB mRNA expression group (43.3%) (P = 0.0199). These results suggest that LACTB may be a prognostic factor for PAAD.

5. it would help strengthen the paper if you could provide a power analysis: that shows the power of your test based on your sample size.

The power of our test has been supplied in table2

---

## [Editor Report · Decision Letter 2]

11 Jan 2021

LACTB mRNA expression is increased in pancreatic adenocarcinoma and high expression indicates a poor prognosis

PONE-D-20-30765R2

Dear Dr. Liu,

We’re pleased to inform you that your manuscript has been judged scientifically suitable for publication and will be formally accepted for publication once it meets all outstanding technical requirements.

Kind regards,

Hiromu Suzuki, M.D., Ph.D.

Academic Editor

PLOS ONE
---

## [Editor Report · Acceptance letter]

13 Jan 2021

PONE-D-20-30765R2 

LACTB mRNA expression is increased in pancreatic adenocarcinoma and high expression indicates a poor prognosis 

Dear Dr. Liu:

I'm pleased to inform you that your manuscript has been deemed suitable for publication in PLOS ONE. Congratulations! Your manuscript is now with our production department. 

Kind regards, 

on behalf of

Dr. Hiromu Suzuki 

Academic Editor

PLOS ONE